# Co-Existence of Certain ESBLs, MBLs and Plasmid Mediated Quinolone Resistance Genes among MDR *E. coli* Isolated from Different Clinical Specimens in Egypt

**DOI:** 10.3390/antibiotics10070835

**Published:** 2021-07-09

**Authors:** Salwa Mahmoud Masoud, Rehab Mahmoud Abd El-Baky, Sherine A. Aly, Reham Ali Ibrahem

**Affiliations:** 1Department of Microbiology and Immunology, Faculty of Pharmacy, Minia University, Minia 61519, Egypt; salwa.mahmoud@mu.edu.eg (S.M.M.); reham.ali@mu.edu.eg (R.A.I.); 2Department of Microbiology and Immunology, Faculty of Pharmacy, Deraya University, Minia 11566, Egypt; 3Department of Medical Microbiology and Immunology, Faculty of Medicine, Assiut University, Assiut 71516, Egypt; s-aly71@windowslive.com

**Keywords:** MDR *E. coli*, ESBLs, MBLs, MAR index

## Abstract

The emergence of multi-drug resistant (MDR) strains and even pan drug resistant (PDR) strains is alarming. In this study, we studied the resistance pattern of *E. coli* pathogens recovered from patients with different infections in different hospitals in Minia, Egypt and the co-existence of different resistance determinants. *E. coli* was the most prevalent among patients suffering from urinary tract infections (62%), while they were the least isolated from eye infections (10%). High prevalence of MDR isolates was found (73%) associated with high ESBLs and MBLs production (89.4% and 64.8%, respectively). *bla_TEM_* (80%) and *bla_NDM_* (43%) were the most frequent ESBL and MBL, respectively. None of the isolates harbored *bla_KPC_* and *bla_OXA-48_* carbapenemase like genes. Also, the fluoroquinolone modifying enzyme gene *aac-(6′)-Ib-cr* was detected in 25.2% of the isolates. More than one gene was found in 81% of the isolates. Azithromycin was one of the most effective antibiotics against MDR *E. coli* pathogens. The high MAR index of the isolates and the high prevalence of resistance genes, indicates an important public health concern and high-risk communities where antibiotics are abused.

## 1. Introduction

*Escherichia coli,* belongs to the family Enterobacteriaceae, is the most common human gastrointestinal commensal as well as important etiological agent of many hospital and community-acquired infections. Pathogenic strains are capable of causing a wide variety of diseases including diarrhea, dysentery, overwhelming sepsis, and the hemolytic-uremic syndrome and neonatal meningitis. *E. coli* can be sorted into intestinal or extraintestinal according to the site of infection [1].

Antibiotics have been the most successful form of chemotherapy developed in the 20th century, saving human lives every day [2]. The evolution of pathogens resistant to antibiotics limits their clinical use, making such infections difficult to control. The antimicrobial resistance (AMR) can be of chromosomal or mobile genetic elements origin [3]. The most common resistance mechanism is the production of the β-lactamase hydrolytic enzymes, which specifically have an inactivated β-lactam ring so that they cannot inhibit the bacterial transpeptidases [4].

β-lactamases are classified into four classes. Serine classes (A, C and D) have serine residue at the hydrolysis active sites. Metallo- β-lactamases (MBLs) (class B) in which the hydrolytic action is promoted by one or two zinc ions at the active site [5]. Class A enzymes include *bla_TEM_* which is the first identified plasmid-encoded β-lactamase; *bla_SHV_* which has similar activity to *bla_TEM_*; *bla_CTX-M_* (cefotaximase) and *bla_KPC_* which confers carbapenem resistance [6]. Class A Extended-spectrum β-lactamase (ESBL) producing strains (*bla_TEM_*, *bla_SHV_* and *bla_CTX-M_* types) are of the most clinically significant pathogens which can resist all β-lactam drugs including monobactams [5,6]. The most clinically significant class B enzymes are *bla_VIM_*, *bla_IMP_* and *bla_NDM_*. MBLs are a group of carbapenemases that resist most β-lactam drugs except the monobactams. Monobactams (e.g., aztreonam) are intrinsically stable to MBLs, but their susceptibility to other serine β-lactamases which are often co-expressed with the MBL limit their usage against MBL expressing strains [7]. Another group that able to hydrolyze carbapenems in addition to other β-lactams are class D β-lactamases (e.g., *bla_OXA-48_* like enzymes) [8]. 

Another example for enzymatic inactivation of antibiotics is the enzymatic modification at different -OH or -NH_2_ groups of aminoglycosides. As a result of the induced steric and/or electrostatic interactions, the modified antibiotic is unable to bind to the target RNA. They can be nucleotidyltransferases (ANTs), phosphotransferases (APHs), or acetyltransferases (AACs) [4]. In addition, the enzyme variant *aac(6′)-Ib-cr* has two amino acid changes that allow the enzyme to inactivate quinolones as well [9,10].

Being plasmid-encoded, hydrolyzing enzymes are likely to be transmissible and widespread. As a single plasmid may encode more than one enzyme, a strain may express many different enzymes, as each one deactivates a different antibiotic [5]. As a result, MDR, or even PDR strains, arising and returning to the pre-antibiotic era has become a nightmare for medical professionals. 

The present study aimed to report the resistance pattern of *E. coli* pathogens, detect the co-existence of different resistance determinants and their correlations to the resistance of *E. coli* pathogens of different infection origins, which would help in identifying local effective therapeutic options and infection control.

## 2. Results

### 2.1. Prevalence of E. coli Among Samples

In the present study, 200 (47%) *E. coli* pathogens were isolated from 425 patients suffering from different infections attending three hospitals in EL-Minia, Egypt. The highest prevalence was among urinary tract infections (62%) while it was lowest among eye infections (10%) (Table 1). Among the three hospitals, *E. coli* isolates were most prevalent in Minia University Hospital samples (51.37%), followed by Minia General Hospital (40.8%) (Table 2).

### 2.2. Antibiotic Resistance of the E. coli Pathogens

The antibiotic susceptibility was tested in 27 antibiotics that cover most of the available antibiotics in the Egyptian market. Appendix A (Appendix A) indicates the different used antibiotics and their different targets. The test revealed that 73% of *E. coli* were MDR. The pathogens were approximately totally resistant to Amoxycillin/clavulanic (97.5%), cephalothin (97%) and cefadroxil (93%). Also, high resistance levels were observed for Ceftazidime (78%) and Aztreonam (65.5%). Imipenem was the most effective antibiotic (20%), followed by Azithromycin (26%) (Figure 1). Our supplementary spread sheet indicates the resistance patterns of the isolates. One hundred MDR isolates were subjected for further investigation. 

### 2.3. Serotyping of the Intestinal E. coli

Out of the selected 100 isolates, 20 were isolated from stool. Since *E. coli* normally inhabit the intestine, stool isolates were serotyped to confirm its pathogenicity. Out of 20 intestinal *E. coli* isolates 15 (75%) isolates were diarrheagenic *E. coli* (DEC). Three isolates (20%) were identified as Enterohaemorrhagic *E. coli* (EHEC) O157:H7. Different O serotypes were observed as O115, O158, O55, O126, O125 and O86a. The identified pathotypes are listed in Appendix A (Appendix A). Untyped five isolates were excluded from the further testing so that 95 isolates were further tested phenotypically and genotypically.

### 2.4. Multiple Antibiotic Resistance Index MDR E. coli Pathogens

The multiple antibiotic resistance index (MARI) ratio between the number of antibiotics that an isolate is resistant to and the total number of antibiotics the organism is exposed to, have been calculated for 95 MDR *E. coli*. It was found that 98.9% of isolates have showed MAR index higher than 0.2, indicating high risk communities where antibiotics are abused. However, there was a statistically significant difference in MARI mean among *E. coli* isolates of different sources (*p* value < 0.05). Eye and blood isolates showed highest MARI mean of 0.82 and 0.74, respectively. On the other hand, stool samples had the lowest MARIs (Figure 2).

### 2.5. Phenotypic Characteristics 

ESBL production phenotypically tested by combined disk test (CDT). It was found that 89.4% (85/95) of the tested strains were ESBL producers. Carbapenemase production was tested by Modified Hodge test (MHT), then carbapenemase producers were tested for MBL using combined-disk synergy test. MBL producers accounted for 50.50% of the isolates (64.8% of carbapenemase producers). Positive significant association was found between ESBL and MBL phenotypes (*p* = 0.001) as all MBL producers were ESBL producers. Statistically significant difference in distribution of ESBL and MBL producers between different infection groups was observed (*p* < 0.001). Regardless of the eye infection, MBL producers were mostly frequent in UTIs (71%) while no intestinal *E. coli* was reported as MBL producers (Table 3).

### 2.6. Antimicrobial Resistance of ESBLs and MBLs Producers

Resistance patterns of ESBL-producers revealed that ESBL producers were highly resistant to β-lactam antibiotics such as amoxycillin/clavulanic (99%), cefadroxil (95%), ceftazidime (85%) and meropenem (69%). MBL producers showed higher resistance rates to same antibiotics (100%, 97.9%, 93.75% and 93.75%, respectively). Azithromycin and chloramphenicol were the most effective drugs against ESBLs and MBLs producers. Appendix A (Appendix A) indicates the antibiogram of ESBLs and MBLs producers.

### 2.7. Prevalence of Resistance Genotypes Among the Tested Isolates

There was a statistically significant difference in the distribution of the different genotypes between different infections. The most prevalent gene was the *bla_TEM_* (80%) followed by *bla_SHV_*, *bla_CTX-M_* and *bla_NDM_* (54.7%, 42% and 44.2%, respectively). All isolates were negative for *bla_KPC_* or *bla_oxa-48_* genes. Also, *the aac-(6′)-Ib-cr* gene was observed in 26.3% of the tested pathogens (Table 4).

### 2.8. Genotypic-Phenotypic Agreement of the Tested Genes

Out of 48 MBL phenotypic positive samples, 38 (79%) isolates were confirmed genotypically as MBL producers. It was found that 15.78% of isolates were phenotypically positive and harbored both *bla_NDM_* and *bla_IMP_* (Figure 3). Furthermore, there was a significant decrease in MAR index when isolates were both *bla_NDM_* and *bla_IMP_* negative (*p* value < 0.001). The *bla_NDM_* producers had higher MAR index than those which were only *bla_IMP_* producers. However, isolates harbored both *bla_NDM_* and *bla_IMP_* was observed resisting higher number of antibiotics (Figure 4). Moreover, positive correlations between MBLs phenotype, genotypes and carbapenem resistance were observed. Statistically significant correlations between detected MBL genotypes and meropenem resistance were observed. The MBLs phenotypes were more significantly associated with bla_NDM_ than bla_IMP_ (Table 5).

ESBLs phenotypes showed strong positive correlation with the presence of *bla_TEM_*. However, all correlations were significant at the 0.01 level. Among the detected ESBL genes, presence of *bla_TEM_* and *bla_CTX-M_* types had the upper hand on the isolates’ resistance followed by *aac(6′)Ib-cr* gene (Table 6).

The current study identified that *aac(6′)Ib-cr* gene was mainly related to aminoglycoside antibiotics than fluoroquinolones. It was found that 47.2% of amikacin resistant isolates and 32.4% of ciprofloxacin resistant isolates harbored *aac(6′)-Ib-cr* gene. The presence of *aac(6′)-Ib-cr* gene was least correlated to ofloxacin resistance. Significant moderate positive correlation was observed between *aac(6′)-Ib-cr* and the resistance to amikacin and tobramycin, *p* values < 0.001 and 0.003, respectively (Table 7).

### 2.9. Association of Different Resistance Genotypes

Most isolates harbored more than one resistance gene (81%). The resistance frequency has significantly increased with the increased number of the co-existed genes (*p* < 0.01). The most frequent association was of the five genes *bla_NDM_*, *bla_IMP_*, *bla_TEM_*, *bla_CTX-M_* and *bla_SHV_* (8.4%) (Table 8).

The correlation matrix of the detected genes indicated overall positive correlations. The strongest and most significant correlation was observed between *bla_CTX-M_* and *bla_SHV_* (r = 0.519). Moreover, aac(6′)Ib-cr gene was significantly associated with *bla_CTX-M_* (Table 9).

Studying the association of *bla_NDM_* gene with class A ESBL genes (*bla_TEM_, bla_CTX-M_* and *bla_SHV_*) among the *bla_NDM_* positive isolates indicated that the association of the *bla_NDM_* with the three ESBLs (*bla_TEM_ + bla_SHV_* + *bla_CTX-M_*) genes was the highest, accounting for 40.47% of the isolates harboring *bla_NDM_* (Figure 5).

Furthermore, the spectrum of antibiotics to which the isolates were resistance is significantly increased (*p* value < 0.01) with the number of positive genes. The isolate harbored five genes showed the highest mean of antibiotic resistance (21 antibiotics), as indicted in Figure 6. As indicated in Appendix A (Appendix A) the resistance of β-lactam drugs is significantly associated with presence of higher number of genes. Also, significant moderate correlation of number of positive genes with ciprofloxacin, norfloxacin and aminoglycoside were observed.

## 3. Discussion 

The last two decades have witnessed a conspicuous increase in the number of infections caused by multi-drug resistant strains of *E. coli*, and this has impacted the outcomes of different infections [11].

The present study demonstrated that the prevalence of *E. coli* pathogens isolated from patients suffering from different infections in El-Minya hospitals accounted for 47%. This result was in accordance with results reported by Amer et al. (45%) [12] and Fam et al. (56%) [13]. 

Among extraintestinal infections, *E. coli* was the most common among urine isolates (62%) followed by blood infections (52%). Likewise, in Saudi Arabia Alanazi et al. [14] reported that *E. coli* was isolated from 60.24% of the urine samples and in Greece Koupetori et al. [15] reported high incidence of *E. coli* accounting for 48% of blood isolates. In contrary, many studies showed lower *E. coli* incidence [16,17]. On the other hand, *E. coli* was least isolated from eye infections (10%). This was higher than results reported in USA by Miller et al. (5.9%) [18].

Moreover, intestinal *E. coli* was serotyped to ensure its pathogenicity. Diarrheagenic *E. coli* (DEC) serotypes accounted for 54.1% which was considered very high in comparison to results obtained by Zhou et al. (7.9%) [19]. *E. coli* O157:H7 accounted for 12.5% of total DEC serotypes. Comparably, in the same region of study Abd El Gany et al. [20] reported incidence rate of 15.72%. The variations between the different studies may be ascribed to many socioeconomical, demographical and geographical factors. It was obviously noted that *E. coli* were relatively high in the present study compared to other studies indicates poor hygienic attitudes that correlated to the mentioned factors. 

Antimicrobial resistance (AMR) emphasizes an overwhelming health and economic burden in both developed and developing countries. As Resistance narrows the therapeutic options leading to increased morbidity and mortality [21]. Our results showed high prevalence of MDR *E. coli* (73%) which were higher than results obtained by Siwakoti et al. [22] and Abdelaziz et al. [23] (28% and 60%, respectively). 

Although carbapenem resistance is considered low, it is higher than previous studies done at the same government [9,20,24]. This may be attributed to the availability and the usage of the drugs when the studies were held. Concerning our results, the drugs were more available and highly used but in previous studies there was a shortage in many antibiotics inside the hospitals. Moreover, the resistance to meropenem was higher than imipenem which may be attributed to that meropenem is cheaper than imipenem, so it is commonly used while some other studies reported complete meropenem sensitivity [9,23].

The continuous spread of ESBLs and carbapenemase mediated resistance has dramatically increased in both hospital and community infections. It was found that 89.4% of the tested isolates were ESBL producers. Abd El-Baky et al. [24] in earlier study in our area reported that 46.8% of isolates were phenotypically ESBL producers. It seems very alerting as the prevalence is almost doubled in short period. 

The incidence of the carbapenemase and MBLs producers accounted for 77.8% and 64.8%, respectively. These rates were very high when compared to results obtained by Ibrahim, et al. [25] who reported that carbapenemase and MBLs incidences were 37.6% and 46.3% respectively. A previous study in our area showed that 52.3% of *P. aeruginosa* were MBL producers [26]. The differences in prevalence may be due to strains and time variations but overall indicate high incidence of MBLs among bacteria in our area. 

There was a significant difference in the distribution of detected genes among the different sample sources *p* values < 0.05 which was in agreement with many studies [27,28]. 

The most prevalent genotype was *bla_TEM_* (80%) followed by *bla_SHV_* (54.7%), *bla_NDM_* (44.2%) and *bla_CTX-M_* (42%). Similarly, *bla_TEM_* was predominant in results reported by Mohamed et al. [28] and Maamoun et al. [29]. On the other hand, a study on *Escherichia coli* causing sepsis among Egyptian children reported that *bla_SHV_* was the most common ESBL (61.22%), followed by *bla_TEM_* (38.78%) and *bla_CTX-M_* (20.41%) [30]. Furthermore, the higher incidence of *bla_TEM_* gene reported by the current study or other studies in our region suggests that *bla_TEM_* gene may be endemic. In contrast to our study, several studies in Asia reported that *bla_CTX-M_* was the most frequent indicating that *bla_CTX-M_* is a predominant genotype in Asia [31,32,33]. Also, reports from Qatar stated that *bla_CTX-M_* type genes evolved through mutations in *bla_TEM_* and *bla_SHV_* genes and it is a recent endemic [34].

Similar to our study, studies in UK have reported 44% of isolates as *bla_NDM_* producers, most of them were from urine samples [35]. The number of *bla_NDM_* producers is increasing in Egypt which is reflected by many studies conducted in this area [30,36,37]. Lower *bla_IMP_* incidence reported in previous studies in Egypt compared to this study (36.8%) suggesting an increasing rate of MBLs producers [36,37]. However, the current reported high prevalence of MBLs may be attributed to the ability of *E. coli* to acquire novel resistance genes through horizontal transfer or the increased use of carbapenems in the clinical treatment. 

*bla_Oxa-48_* like and *bla_KPC_* were not detected in any isolate. Quite higher prevalence of *bla_Oxa-48_* like and *bla_KPC_* (38.46% and 23%, respectively) was reported in Bangladesh [38]. In accordance with the current study, *bla_KPC_* wasn’t detected in several previous studies in Egypt or detected in very low rate [37,38]. In addition, *bla_KPC_* wasn’t detected in countries such as Saudi Arabia [39] or those of the Arabian Peninsula [40]. These data confirmed that *bla_KPC_* genes does not predominate in this geographical region, where it is frequently detected in the United States [41] and endemic in Israel [42]. The *aac(6′)-Ib-cr* gene prevailed in 26.3% of the isolates which were mostly isolated from wounds. This rate is lower than rates previously reported by Al-Agamy et al. [43] and Mohamed et al. [28]. The differences across studies may be attributed to differences in geographical locations, age groups, or clinical criteria. 

Most of the isolates (81%) harbored more than one resistance determinant. Co-harboring of multiple ESBL genes was detected previously in Egypt [28,44] and some other countries; Burkina Faso [45], Qatar [33] and Iran [46].The co-existence of *bla_NDM_, bla_IMP_, bla_TEM_, bla_CTX-M_* and *bla-_SHV_* was the most frequent, accounting for 8.5% of the isolates. There was a significant association between *bla_CTX-M_* and *bla_SHV_*, which agree with other studies [28,30,46]. There was a significant positive correlation between *bla_NDM_* and *bla_IMP_*. This was comparable to Zaki et al. [30] and Kamel et al. [47] where single *E. coli* isolate had more than one type of metallo β-lactamase. The association between *aac(6′)-Ib-cr* and *bla_CTX-M_* genes was statistically significant, agreeing with previous studies [48,49,50]. None of the isolates harbored *aac(6′)Ib-cr* alone. Moreover, there was significant association of *aac(6′)-Ib-cr* gene with ESBL phenotypes. This may be due to the common presence of ESBL genes and PMQR genes on the same plasmid in *Enterobacteriaceae* [51]. Moreover, the *aac(6′)-Ib-cr* gene showed significant positive correlation with amikacin and tobramycin resistance in ESBL producers. Similarly, Mohamed et al. [28] reported significant association of ESBL genes with *aac(6′)-Ib-cr* gene that resulted in increased ciprofloxacin, gentamicin and amikacin resistance in ESBL producers.

The resistance rates were significantly increased in ESBL producers than non-producers that reported by several studies [28,52,53,54]. In agreement with our results, many studies reported higher resistance rate of MBL producers in comparison to MBLs non-producers. In accordance with the current work, previous studies have reported significant high resistance rates in MBL producers [55,56]. It was reported that *bla_TEM_* and *bla_SHV_* are important factors in increased resistance of ESBL *E. coli* producers to third-generation cephalosporin [57]. 

Finally, variations in rate and predominance of resistance genes between different countries and even among the same country institutions may be due to difference in locally prescribed antibiotics and if the infection control guidelines are followed or not in different health institutes. In Egypt, the high rate of ESΒLs and MBLs is a reflection of the inappropriate use of antimicrobials due to the over counter availability of antibiotics without prescription and patients incompliance or the wide use of antibiotics in veterinary care and farms [58,59].

## 4. Material and Methods

### 4.1. Bacterial Isolates 

Two hundred *E. coli* isolates were isolated from 425 patients attending different hospitals in El-Minia with different infections. All clinical samples were obtained as part of the routine hospital laboratory procedures. Samples were processed and cultured on trypticase soy agar (Lab M, Hewwood, UK) at 37 °C for overnight. *E. coli* colonies gave pink color on MacConkey agar and green metallic sheen on Eosin methylene blue (EMB) (lab M, Hewwood, UK). Colonies were further identified by regular microbiological biochemical tests [60].

### 4.2. Antimicrobial-Susceptibility Testing

The antimicrobial susceptibility of the isolates was tested by the Kirby-Bauer Disk Diffusion method [61]. The used antibiotics discs were ready cartilages purchased from Oxoid; Basingstoke, UK. The following antibiotic discs were used Cefpodoxime (10 μg), Streptomycin (10 μg), Aztreonam (30 μg), Ceftriaxone (30 μg), Gentamycin (10 μg), Amoxycillin/clavulanic (20/10 μg), Piperacillin/Tazobactam (100/10 μg), Ceftazidime (30 μg), Imipenem (10 μg), Meropenem (10 μg), Cefoperazone (75 μg), Doxycycline (30 μg), Ciprofloxacin (5 μg), Amikacin (30 μg), Nalidixic acid (30 μg), Cefotaxime (30 μg), Cefepime (30 μg), Ampicillin/sulbactam (10/10 μg), Norfloxacin (10 μg), Tobramycin (10 μg), Sulfamethoxazole/trimethoprim (23.75/1.25 μg), Chloramphenicol (30 μg). Isolates classified as sensitive, intermediate and resistant according to inhibition zones interpretation standards of Clinical Laboratory standards Institute (CLSI) 2018 (Appendix A) [62].

### 4.3. Serotyping of Intestinal E. coli

*Escherichia coli* recovered from gastroenteritis infections were sent to the Animal Health Research Institute, Giza, Egypt to be serotyped. The isolates serotyped through detection of isolates agglutination with O and H antisera using the slide agglutination method according to the manufacturer instructions (Pro-Lab Diagnostics, Round Rock, TX, USA).

### 4.4. Phenotypic Detection of ESBLs and MBLs Production

Detection of ESBL in *E. coli* isolates was carried out by combined disc test (CDT). Isolates defined positive when the difference between the inhibition zones of cefotaxime and cefotaxime/clavulanic or ceftazidime and ceftazidime/clavulanic disks is ≥5 mm [62]. Carbapenemases were detected in carbapenem resistant isolates by Modified Hodge test (MHT). MHT positive isolates w ere further tested for MBL production using EDTA-combined disk synergy test. An increase in zone diameter of at least 7 mm around the imipenem–EDTA or meropenem–EDTA disks were recorded positive result [63].

### 4.5. Amplification of Resistance Genes

The DNA templet was extracted by available commercial kit QIAprep^®^ Spin Miniprep Kit (QIAGEN, Germany) by following the manufacturer instruction. Resistance genes were detected using conventional PCR technique. Amplification was done using 25 μL PCR reaction mixture consisting of 12.5 μL master mix (BIOMATIk, Kitchener, Canada) 1 μL of each forward and reverse primers (BIOMATIk, Canada), 2 μL DNA template and 8.5 μL nuclease-free water. PCR cycling conditions are indicated in Appendix A (Appendix A).

### 4.6. Statistical Analysis

Data were analyzed using IBM SPSS version 20.0. First, normal distribution of data was tested by normality tests as Kolmogorov–Smirnov and Shapiro–Wilk *p*-values in addition to histograms. Descriptive analysis was done to analyze prevalence of *E. coli* isolates among different infections and hospitals, percentage of resistance and prevalence of ESBL and MBL producers and prevalence of the different genes. To compare differences in distribution between different groups chi-square (X^2^) test was done but when more than 20% of cells were less than 5, Fisher’s exact test was done to be more accurate. One-way ANOVA tests was done to compare mean values between different groups as MAR index mean values in different sample sources. Non parametric tests were used for non-parametric data as Kruskal–Wallis. To study associations between phenotypes, genotypes and resistance, correlations were established using Pearson’s correlation coefficient (r^2^) in bivariate. *p*-values are significant if they are ≤0.05.

## 5. Conclusions

High resistance reported in our study indicates poor awareness of the microbiological laboratory test importance, high empirical antimicrobial prescription and high patient incompliance. Moreover, the massive co-existence of the detected genes strongly supports the presence of one or more circulating plasmids that harbors different resistance genes. Finally, the study highlighted the importance of continuous surveillance of the resistance trends and the direct need to strictly apply the infection control policies, implementing a national antimicrobial stewardship plan.

## Figures and Tables

**Figure 1 antibiotics-10-00835-f001:**
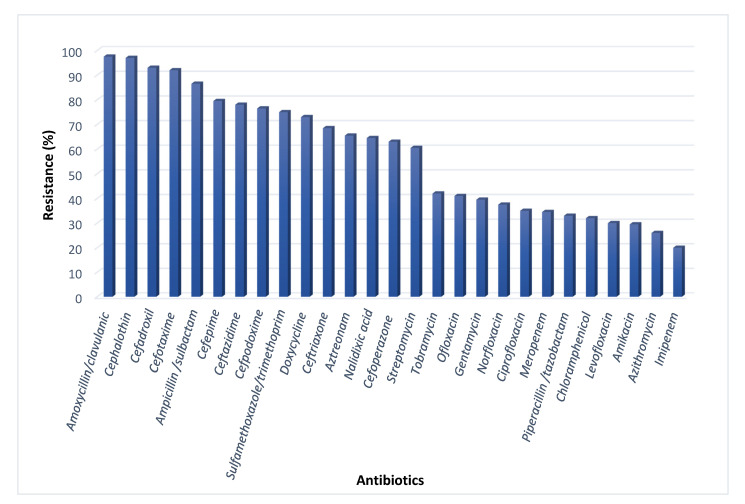
Antibiotic resistance of the total *E. coli* isolates.

**Figure 2 antibiotics-10-00835-f002:**
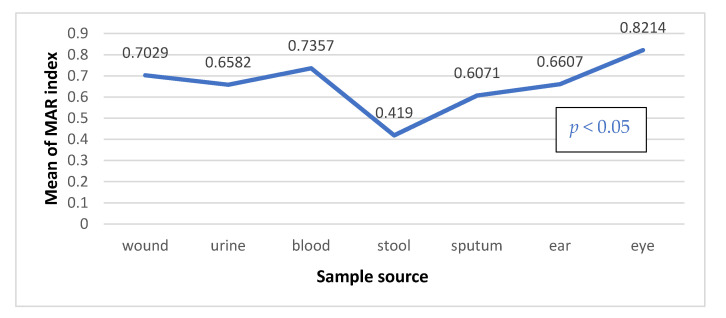
MAR index mean values ($\bar{x}$), *p* value calculated by One-way ANOVA test.

**Figure 3 antibiotics-10-00835-f003:**
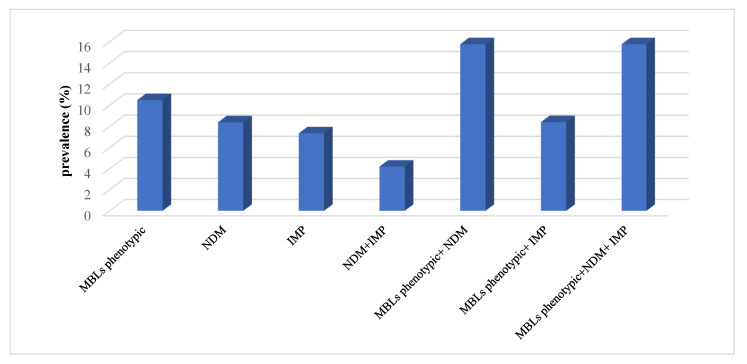
Phenotypic and genotypic agreement of MβL tested genes.

**Figure 4 antibiotics-10-00835-f004:**
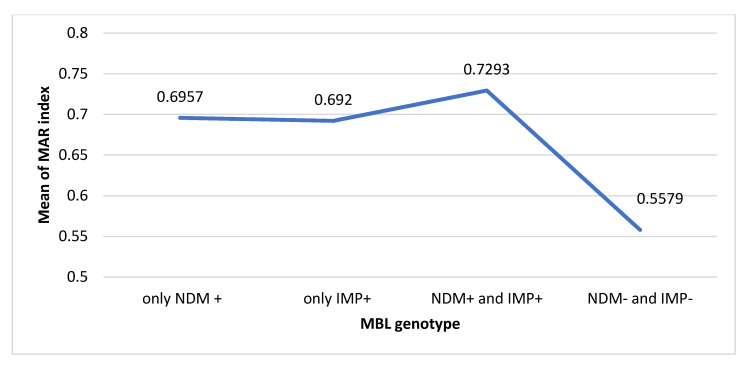
Distribution of MAR index mean values ($\bar{x}$) among the detected MBLs genotypes.

**Figure 5 antibiotics-10-00835-f005:**
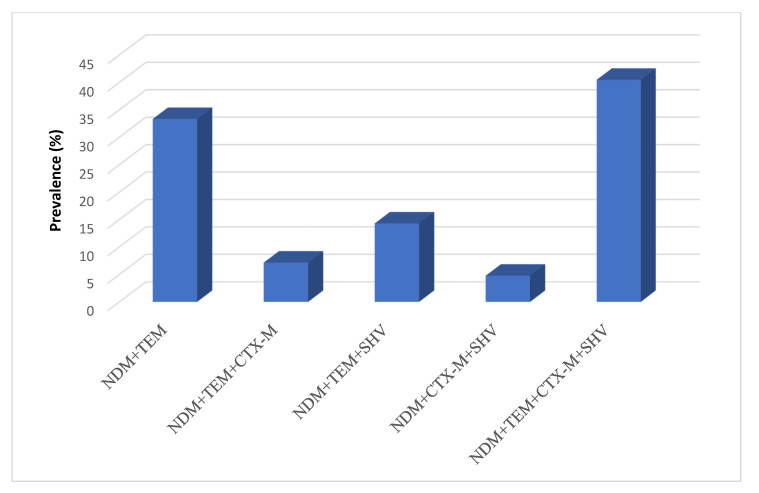
*bla**_NDM_* association with class A ESBLs genes.

**Figure 6 antibiotics-10-00835-f006:**
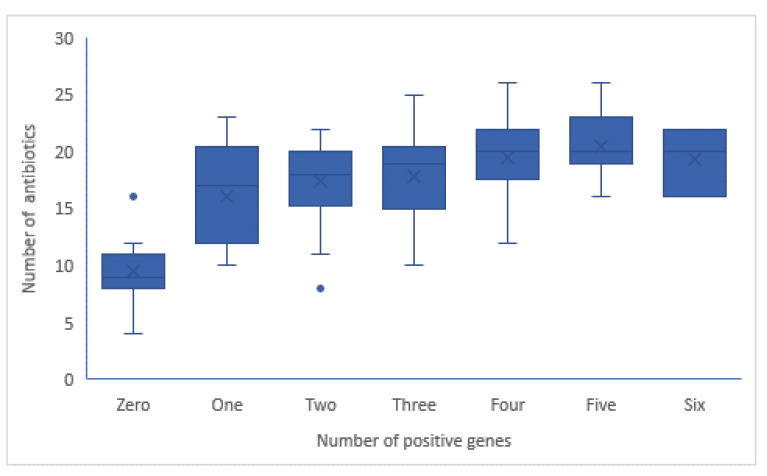
Compatibility of number of detected genes and number of antibiotics to which isolates were resistant among 95 isolates.

**Table 1 antibiotics-10-00835-t001:** Prevalence of *E. coli* among different clinical samples.

Infection	No. of Samples	No. of *E. coli* Isolates	*E. coli* (%) *
Wound infections (burns, diabetic foot, surgery wound, cuts)	150	66	44%
Urinary tract infection	100	62	62%
Gastro-enteritis	50	24	48%
Blood	75	39	52%
Chest infection	20	4	20%
Ear infection	20	4	20%
Eye	10	1	10%
Total	425	200	47%

* Percent of *E. coli* were correlated to the number of samples of each infection.

**Table 2 antibiotics-10-00835-t002:** Distribution of *E. coli* among samples collected from different hospitals.

Hospitals	No. of Samples	*E. coli*	
		No.	% *
Minia University Hospitals	290	149	51.37
Minia Chest Hospital	20	4	20
Minia General Hospital	115	47	40.8
Total	425	200	

* Percent of *E. coli* were correlated to the number of collected samples from each hospital.

**Table 3 antibiotics-10-00835-t003:** Distribution of ESBLs and MBLs producing isolates among clinical specimens.

Type of Infection	β-Lactam Resistant Isolates	ESBLs		MBL	
		No.	% *	No.	% *
Wound infections	25	25	100%	12	48%
Urinary tract infection	28	28	100%	20	71.4%
Gastro-enteritis	15	5	33.3%	0	0%
Blood	20	20	100%	12	46.67%
Respiratory infection	4	4	100%	2	50%
Ear infection	2	2	100%	1	50%
Eye	1	1	100%	1	100%
Total	95	85	89.4% **	48	50.5% **

* percent was correlated to the total number of samples of each infection type; ** percent was correlated to total number of samples. Significant if *p* ≤ 0.05.

**Table 4 antibiotics-10-00835-t004:** Distribution of detected MBLs and ESBLs genotypes.

Type of Infection	β-Lactam Resistant Isolates N	*bla_NDM_*	*bla_TEM_*	*bla_CTX-M_*	*bla_SHV_*	*bla_IMP_*	*aac-(6′)-Ib-cr*
		N (%) *	N (%) *	N (%) *	N (%) *	N (%) *	N (%) *
Wound infections	25	14 (56%)	23 (92%)	14 (56%)	15 (60%)	12 (48%)	9 (36%)
UTI	28	17 (60.7%)	27 (96.4%)	11 (39.2%)	12 (42.8%)	16 (57%)	6 (21%)
Gastro-enteritis	15	1 (7.6%)	7 (46.6%)	0 (0%)	0 (0%)	0 (0%)	0 (0%)
Blood	20	7 (35%)	20 (100%)	13 (65%)	20 (100%)	7 (35%)	10 (50%)
Chest infection	4	1 (25%)	3 (75%)	0 (0%)	4 (100%)	0 (0%)	0 (0%)
Ear infection	2	1 (50%)	2 (100%)	1 (50%)	0 (0%)	0 (0%)	0 (0%)
Eye	1	1 (100%)	1 (100%)	1 (100%)	1 (100%)	0 (0%)	0 (0%)
Total	95	42 (44.2%)	76 (80%)	40 (42%)	52 (54.7%)	35 (36.8%)	25 (26.3%)
*p* value **		0.11	<0.001	0.011	<0.001	0.011	0.042

* percent was correlated to the total number of β-lactam resistant isolates in each type of infection. ** Significant *p* value at *p* ≤ 0.05.

**Table 5 antibiotics-10-00835-t005:** Relation between MBLs phenotype, detected genotype and carbapenem resistance across the *E. coli* isolates.

	MBLs Phenotype	*bla_NDM_*	*bla_IMP_*	Imipenem Resistance	Meropenem Resistance
MBLs phenotype	1	0.372 **	0.232 *	0.237 *	0.465 **
*bla_NDM_*		1	0.155	0.054	0.212 *
*bla_IMP_*			1	0.122	0.275 **
Imipenem resistance				1	0.292 **
Meropenem resistance					1

* Correlation is significant at the 0.05 level (2-tailed). ** Correlation is significant at the 0.01 level (2-tailed). *p* values calculated by Fisher’s exact test.

**Table 6 antibiotics-10-00835-t006:** Correlation matrix (r^2^) between phenotypes, genotypes and antibiotic resistance across the *E. coli* isolates.

	ESBLs Production	MBLs Production	*bla_NDM_*	*bla_IMP_*	*bla_TEM_*	*bla_CTX-M_*	*bla_SHV_*	*aac(6′)1b-cr*
MAR index	0.611 **	0.342 **	0.330 **	0.289 **	0.366 **	0.365 **	0.251 *	0.360 **
ESBLs phenotype	1	0.347 **	0.305 **	0.262 *	0.696 **	0.293 **	0.377 **	0.205 *

* Correlation is significant at the 0.05 level (2-tailed). ** Correlation is significant at the 0.01 level (2-tailed). *p* values calculated by Fisher’s exact test.

**Table 7 antibiotics-10-00835-t007:** Correlation between *aac(6′)Ib-cr*, aminoglycoside and fluoroquinolone resistance.

Antibiotics	Number of Resistant Isolates	Number of aac(6′)Ib-cr Positive Isolates (%) *	Person Correlation (r^2^)	*p* Value
Streptomycin	61	19 (31)	0.135	0.193
Tobramycin	43	17 (39.5)	0.301	0.003 **
Gentamycin	43	13 (30.2)	0.044	0.670
Amikacin	36	17 (47.2)	0.374	<0.01 ***
Ofloxacin	39	9 (23)	0.018	0.866
Norfloxacin	42	14 (33.3)	0.1450	0.162
Ciprofloxacin	37	12 (32.4)	0.152	0.142

* percent correlated to no. of resistant isolates of each antibiotic. *p* values were calculated by Fisher’s exact test. ** *p* value is significant at 0.05 level (2-tailed), *** *p* value is significant at 0.01 level (2-tailed).

**Table 8 antibiotics-10-00835-t008:** Co-existence of different genotypes.

	Number of Isolates (%)
One gene	9 (9.5)
*bla_TEM_* *bla_SHV_*	8 (8.5) 1 (1)
Two genes	20 (21.1)
*blaTEM, aac(6′)Ib-cr* *blaIMP, blaTEM* *blaTEM, blaSHV* *blaNDM, blaTEM* *blaTEM, blaCTX-M*	1 (1) 5 (5.2) 5 (5.2) 7 (7.3) 2 (2.1)
Three genes	21 (22.1)
*blaIMP, blaTEM, blaSHV* *blaIMP, blaTEM, blaCTX-M* *blaNDM, blaTEM, aac(6′)Ib-cr* *blaTEM, blaCTX-M, blaSHV* *blaTEM, blaSHV, aac(6′)Ib-cr* *blaNDM, blaCTX-M, blaSHV* *blaNDM, blaIMP, blaTEM* *blaNDM, blaTEM, blaSHV*	4 (4.2) 1 (1) 2 (2.1) 5 (5.2) 1 (1) 1 (1) 2 (2.1) 5 (5.2)
Four genes	21(22.1)
*blaTEM, blaCTX-M, blaSHV, aac(6′)Ib-cr* *blaNDM, blaIMP, blaTEM, aac(6′)Ib-cr* *blaNDM, blaTEM, blaCTX-M, blaSHV* *blaIMP, blaTEM, blaCTX-M, blaSHV* *blaIMP, blaTEM, blaSHV, aac(6′)Ib-cr* *blaNDM, blaIMP, blaTEM, blaCTX-M* *blaNDM, blaTEM, blaSHV, aac(6′)Ib-cr* *blaNDM, blaCTX-M, blaSHV, aac(6′)Ib-cr* *blaNDM, blaTEM, blaCTX-M, aac(6′)Ib-cr*	5 (5.2) 3 (3.1) 3 (3.1) 3 (3.1) 1 (1) 1 (1) 1 (1) 1 (1) 3 (3.1)
Five genes	12 (12.6)
*blaNDM, blaIMP,, blaTEM, blaCTX-M, blaSHV* *blaNDM, blaIMP, blaTEM, blaCTX-M, aac(6′)Ib-cr* *blaIMP, blaTEM, blaCTX-M, blaSHV, aac(6′)Ib-cr*	8 (8.5) 2 (2.1) 2 (2.1)
Six genes	3 (3.2)
Total	95

Percentages were correlated to the total number of isolates.

**Table 9 antibiotics-10-00835-t009:** Correlation matrix (r^2^) between the different genotypes.

	*bla_NDM_*	*bla_IMP_*	*bla_TEM_*	*bla_CTX-M_*	*bla_SHV_*	*aac(6′)1b-cr*
***bla_NDM_***	1	0.155	0.211 *	0.185	0.086	0.190
***bla_IMP_***		1	0.290 **	0.233 *	0.081	0.089
***bla_TEM_***			1	0.196	0.227 *	0.155
***bla_CTX-M_***				1	0.519 **	0.265 **
***bla_SHV_***					1	0.159
***aac(6′)Ib-cr***						1

* Correlation is significant at the 0.05 level (2-tailed). ** Correlation is significant at the 0.01 level (2-tailed). *p* values were calculated by Fisher’s exact test.

## Data Availability

Not applicable.

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
