# Peer review of "Co-Existence of Certain ESBLs, MBLs and Plasmid Mediated Quinolone Resistance Genes among MDR E. coli Isolated from Different Clinical Specimens in Egypt"

_antibiotics, 2021, doi:10.3390/antibiotics10070835_

Round 1
Reviewer 1 Report
In the present manuscript, the Authors identified wide presence of E. coli in patients from three different hospitals in Egypt. Out of 425 samples, E. coli was identified in 200 samples. Further, resistance to several antibiotics was measured by the Kirby-Bauer Disk Diffusion method, where samples from urinary tract infections had the highest presence of E. coli (62%). Later serotyping was also performed. Overall, this manuscript uses traditional microbiology methods to identify the presence of antibiotic resistant E. coli. I do not find any exciting or novel findings in this manuscript. Here are my specific comments:
- Introduction section is poorly written. Authors focus only on the antibiotic resistance mediated by beta-lactamase, while they measured resistance on a diverse range of antibiotics such as antibiotics inhibiting translation.
- Samples were collected from different sites but, data showing the distribution of antibiotic resistance across these sites are missing. Please include a supplementary excel file showing raw data including, the name of the site, type of disease or body parts from where samples were collected. Based on this data, included error bar in Figure 2. Also, if you follow these steps, Figure 1 can be removed.
- Please mention how many antibiotics were used? How were they chosen? Please make a supplementary table describing different biological processes targeted by these antibiotics such as cell wall, translation, transcription, and replication.
- Please mention why only 100 samples were chosen for serotyping? Were samples picked up randomly or there was any logic behind that?
- Please describe in detail how E. coli strains were classified as sensitive, intermediate, and resistant. Describe how different thresholds on inhibition zones were decided? Include the supplementary file containing the raw data.
- It is not evident if the resistance of every E. coli strain was measured against all antibiotics? Please include a supplementary file describing MDR profiles of each strain.
- It is hard to understand how p-values were calculated in table 2. Also move table 1, 3, 7 and 10 to supplementary.
- I feel that several figures can be combined, and quality can be improved.
- Authors selected 200 E. coli, but in several cases, measurement was done only with 95 strains. For example, analysis done in tables 2 & 4.
- The manuscript has tons of typos. For example, references line 83-85, 93-95, and all references.
- The result section is superficially written and boring. At the beginning of each section authors should mentions why are they performing that task and what is significance. Also, there is a huge gap in information flow between different sections.
- Y- axis label is missing in Figure 4. Although data presented there seems to be MAR index but the Figure legend mentions variance? Why do Authors talk about MAR index >0.2? Is there any statistical significance? Please describe, how variance and p-values were calculated.
- How MAR indexes were correlated among three different sites (hospitals). Please calculate at the level of the organ such as blood, stool, ear, eye, etc.
Author Response
# Reviewer 1
Comment 1:
Introduction section is poorly written. Authors focus only on the antibiotic resistance mediated by beta-lactamase, while they measured resistance on a diverse range of antibiotics such as antibiotics inhibiting translation.
Response to Comment 1:
Thank you for this comment….
- In our study we tested the resistance of coli isolates against a wide range of antibiotics that commonly used in the Egyptian market. On the other hand, the further of the study focused on the detection of resistance mediated by beta-lactamase in our region phenotypically, genotypically and statistically hence the introduction focused on it.
Comment 2
Samples were collected from different sites but, data showing the distribution of antibiotic resistance across these sites are missing. Please include a supplementary excel file showing raw data including, the name of the site, type of disease or body parts from where samples were collected. Based on this data, included error bar in Figure 2. Also, if you follow these steps, Figure 1 can be removed.
Response to Comment 2:
Thank you for this comment…
- The required excel sheet is provided in the supplementary data. To make data in figures 1 and 2 more clear, the figures are replaced by tables 1 and 2. They are highlighted using track changes function in the text.
Comment 3:
Please mention how many antibiotics were used? How were they chosen? Please make a supplementary table describing different biological processes targeted by these antibiotics such as cell wall, translation, transcription, and replication.
Response to Comment 3:
Thank you for this comment.
- In our study we used 27 antibiotics to cover most available antibiotics in the Egyptian market. Moreover, hospitals from which samples were collected are of the largest hospitals in our area and a large number of patients from different environments attending them so we tried to cover different antibiotics regimen they were probably exposed to before and during their stay in the hospital. Table S1 (supplementary) indicates the targeted process of the tested antibiotics.
Comment 4:
Please mention why only 100 samples were chosen for serotyping? Were samples picked up randomly or there was any logic behind that?
Response to Comment 4:
Thank you for this comment.
- In our study total 200 coli were isolated, 100 MDR E. coli were selected randomly for the further investigations. Out of the 100 isolates, 20 intestinal isolates only were serotyped to confirm their pathogenicity (not commensal). Out of 20 isolates only 15 were typed. The 5 untyped isolates were excluded from further analysis making total analyzed pathogens are 95.
Comment 5:
Please describe in detail how E. coli strains were classified as sensitive, intermediate, and resistant. Describe how different thresholds on inhibition zones were decided? Include the supplementary file containing the raw data.
Response to Comment 5:
Thank you for this comment.
- The coli isolates were classified to sensitive, Intermediate and resistant according to the inhibition zones interpretation standards of the Clinical Laboratory standards Institute (CLSI) 2018.
- As recommended, Table S2 (supplementary) shows the different standard inhibition Zones.
Reference
Wayne, P. CLSI. Performance Standards for Antimicrobial Susceptibility Testing. CLSI supplement M100. Clinical and Laboratory Standards Institute 2018, 28th ed.
Comment 6:
It is not evident if the resistance of every E. coli strain was measured against all antibiotics? Please include a supplementary file describing MDR profiles of each strain.
Response to Comment 6:
Thank you for this comment.
- The resistance of all coli isolates was tested against all tested antibiotics (27). The supplementary excel file shows the different sites of isolates, the sample type and the sensitivity test results of each isolate.
Comment 7:
It is hard to understand how p-values were calculated in table 2. Also move table 1, 3, 7 and 10 to supplementary.
Response to Comment 7:
Thank you for this comment.
- The difference of distribution of ESBL and MBL between different infection groups were tested by Fisher’s exact test, there was an overall significant difference between groups. This significance can is mainly due to the difference between intestinal and extraintestinal infections where prevalence is higher in extraintestinal ones.
- Pearson’s correlation coefficient (r2) used to detect association of ESBLs and MBLs, it was statistically significant as all MBL producers were also ESBL producers.
- As recommended tables 1,3,7 and 10 are moved to supplementary data.
Comment 8:
I feel that several figures can be combined, and quality can be improved.
Response to Comment 8:
Thank you for this comment.
- The figures are checked for any possible combinations
- Figures I and 2 are replaced by tables.
- Figure 7 and Table 7 are combined in single table (Table7).
Comment 9:
Authors selected 200 E. coli, but in several cases, measurement was done only with 95 strains. For example, analysis done in tables 2 & 4.
Response to Comment 9:
Thank you for this comment.
- In our study we isolated total 200 coli isolate from 425 patient. Sensitivity tests were done for all the isolates. then, 100 MDR isolate from different infections were selected randomly. Out of the 100 isolates, 20 isolates were intestinal so they were serotyped to confirm their pathogenicity (not commensal). Five untyped isolates were excluded from the further investigations so that the number of strains become 95.
Comment 10:
The manuscript has tons of typos. For example, references line 83-85, 93-95, and all references.
Response to Comment 10:
Thank you for this comment.
- As recommended, we edited the manuscript and revised it and typos is corrected.
Comment 11:
The result section is superficially written and boring. At the beginning of each section authors should mentions why are they performing that task and what is significance. Also, there is a huge gap in information flow between different sections.
Response to Comment 11:
Thank you for this comment.
- The result section is revised and modified as possible. For example, figure 1 and 2 are replaced by tables to represent the data more clearly. Also, some sections are handled in more details to fill the unintended gap in the information flow.
Comment 12:
Y- axis label is missing in Figure 4. Although data presented there seems to be MAR index but the Figure legend mentions variance? Why do Authors talk about MAR index >0.2? Is there any statistical significance? Please describe, how variance and p-values were calculated.
Response to Comment 12:
Thank you for this comment.
- The missed Y axis label is fixed.
- The MARI >0.2 is mentioned as it is indicating high risk communities where antibiotics are abused.
- P- values is calculated using one-way ANOVA tests to compare MARI between different groups and variance is tested within the ANOVA test to test homogeneity of the groups. The ANOVA test showed overall statistically significant difference, post hoc tests that significance is mainly comes from difference between intestinal and extra-intestinal isolates where extra-intestinal groups showed higher resistance than the intestinal one.
- The word variance in the figure 4 caption is removed as the chart plotting the MARI mean values of the different infection groups.
Comment 13:
How MAR indexes were correlated among three different sites (hospitals). Please calculate at the level of the organ such as blood, stool, ear, eye, etc.
Response to Comment 13:
Thank you for this comment.
- The MAR indexes in figure 4 were calculated for the selected MDR 95 isolates at the level of the organ
- MAR indexes can be correlated to the different hospitals for the 95 isolates as follows
- And for the total number of isolated coli (200 isolates) as follows:
Reviewer 2 Report
Antibiotic resistance is a growing global menace. In this study, Masoud et al. report the resistance pattern of E.coli pathogens isolated from over 400 patients in Egypt's hospitals. This article would be of broad interest to readers of the antibiotics journal. However, some items need to be addressed before acceptance:
- My primary concern is the lack of relationship analysis between the co-existence of different beta-lactamase genes and the resistance profile of the pathogens. The authors evaluate the resistance patterns of the E.coli isolated using disc diffusion assay and characterize the genotype of the isolates using PCR. However, it is essential to illustrate their correction in this study. For instance, does the co-existence of five or six genes in Table 8 causes higher resistance for certain beta-lactam types or provide a broader spectrum for different antibiotics?
- In addition, the authors could strength the manuscript by further exploring the beta-lactamase gene identified in this project. For example, Soliman et al. report the Emergence of an NDM-5-producing clinical Escherichia coli isolate in Egypt. It would be interesting to know if the beta-lactamase gene identified in this study is WT or other clinical variants by sequencing the PCR product.
- Finally, authors must improve the format of the manuscript by following the "instructions to authors file" provided by the journal.
- All Figures and Tables should be inserted into the main text close to their first citation and must be numbered following their number of appearances. For example, authors need to switch Figure 1 and Figure 2 on Page 3 and add "(Figure 1)" and "(Figure 2)" to lines 72-77.
- It would be nice for authors to correct the reference numbers in square brackets. Many of them show "Error! Reference source not found".
- Authors need to reformat the reference according to "Instructions to Authors" provided by the Antibiotic journal.
- authors need to change the font size for "Samples were." Line 302 and “Enterobacteriaceae” on line 279.
Author Response
#Reviewer 2:
Comment 1:
My primary concern is the lack of relationship analysis between the co-existence of different beta-lactamase genes and the resistance profile of the pathogens. The authors evaluate the resistance patterns of the E. coli isolated using disc diffusion assay and characterize the genotype of the isolates using PCR. However, it is essential to illustrate their correction in this study. For instance, does the co-existence of five or six genes in Table 8 causes higher resistance for certain beta-lactam types or provide a broader spectrum for different antibiotics?
Response to Comment 1:
Thank you for this comment.
- The number of antibiotics to which isolates are resistant is increased by the number of positive genes, groups that have 4 and 5 genes were the highest, Figure 6 is added indicating the mean, maximum and minimum number of resistant antibiotics of each group. Also, Table S5 indicates the person correlation of the number of harbored genes with each tested antibiotic resistance and the MAR index is included supplementary data.
Comment 2:
In addition, the authors could strength the manuscript by further exploring the beta-lactamase gene identified in this project. For example, Soliman et al. report the Emergence of an NDM-5-producing clinical Escherichia coli isolate in Egypt. It would be interesting to know if the beta-lactamase gene identified in this study is WT or other clinical variants by sequencing the PCR product.
Response to Comment 2:
Thank you for this comment.
- Further exploring of the clinical variants of detected genes in our study would of course strength our manuscript, it is a limitation in our study. Unfortunately, the sequencing was beyond the financial capacity of the researchers as the project was self-funded and we didn’t receive any external funding.
Comment 3:
Finally, authors must improve the format of the manuscript by following the "instructions to authors file" provided by the journal.
- All Figures and Tables should be inserted into the main text close to their first citation and must be numbered following their number of appearances. For example, authors need to switch Figure 1 and Figure 2 on Page 3 and add "(Figure 1)" and "(Figure 2)" to lines 72-77.
- It would be nice for authors to correct the reference numbers in square brackets. Many of them show "Error! Reference source not found".
- Authors need to reformat the reference according to "Instructions to Authors" provided by the Antibiotic journal.
- authors need to change the font size for "Samples were." Line 302 and “Enterobacteriaceae” on line 279.
Response to Comment 3:
Thank you for this comment.
As recommended the manuscript is revised and edited.
- Figure 1 and 2 are removed and tables are added instead.
- The order of tables and figures is revised and modified.
- Regarding the references we used the MDPI endnote style that provided by the journal.
- The font size for "Samples were." Line 302 and “Enterobacteriaceae” on line 279 is corrected
Round 2
Reviewer 1 Report
The authors have made significant changes in the manuscript. But, the manuscript still has significant number of typos. For example: At several places, species have not been written in italic, choose between "percentage" or "%", Y-labels of figures 3 & 4 are missing. Please re-review the full manuscript and fix the typos.
Author Response
Comment 1:
The authors have made significant changes in the manuscript. But, the manuscript still has significant number of typos. For example: At several places, species have not been written in italic, choose between "percentage" or "%", Y-labels of figures 3 & 4 are missing. Please re-review the full manuscript and fix the typos.
Response to Comment 1:
Thank you for this comment….
- The manuscript is revised and detected typos are fixed
- Species are checked and turned italic
- We changed all percentage to %
- Y-labels in figures 3 &4 are added

Reviewer 2 Report
The authors have address my concerns. The revised manuscript looks good.
Author Response
Thanks for your great effort